# An Update on Vitamin D Deficiency Status in Malaysia

**DOI:** 10.3390/nu14030567

**Published:** 2022-01-27

**Authors:** Zaleha Md Isa, Nor Rumaizah Mohd Nordin, Muhammad Hilmi Mahmud, Syahirah Hashim

**Affiliations:** Department of Community Health, Faculty of Medicine, National University of Malaysia, Jalan Yaacob Latif, Bandar Tun Razak, Cheras, Kuala Lumpur 56000, Malaysia; rumaizahnordin@gmail.com (N.R.M.N.); p108904@siswa.ukm.edu.my (M.H.M.); syahirahhashim26@gmail.com (S.H.)

**Keywords:** vitamin D deficiency, prevalence, Malaysia

## Abstract

Vitamin D is essential for maintaining serum calcium levels, ensuring sufficient bone mineralization, immunomodulatory properties, and a protective effect on the cardiovascular system, renal disease, cancer, as well as in pregnancy. Vitamin D deficiency is prevalent worldwide, and it is not related to a country’s development index. However, the data on vitamin D deficiencies are primarily taken from out-of-date, small-scale studies on target age groups or specific diseases, rather than from large-scale, population-based surveys. In Malaysia, for the past 16 years, studies were conducted involving adult men and women, pregnant women, postmenopausal women, adolescent, and children especially with specific diseases such as spina bifida, epilepsy, chronic liver disease, and atopic dermatitis. Only a few large surveys were conducted involving children and adolescents. Across the specific target population studied, vitamin D deficiency and insufficiency were seen particularly among females, Indians, and those of Malay ethnicity. This is related to widely known causes of vitamin D deficiency such as skin type (melanin) and sun avoidant lifestyles that include covering clothes, largely practiced by Malay Muslims in Malaysia. Other related causes or the high-risk groups are breastfed infants, the elderly, the obese, those on medications, and those characterized by fat malabsorption and geophysical factors. Vitamin D deficiency can be managed with pharmacological or non-pharmacological approaches, depending on the severity. The objective is to raise serum vitamin D to a normal level, hence, relieving the symptoms and reducing the adverse health outcomes. Despite no clear guidelines in treating vitamin D deficiency in Malaysia, this condition can be prevented with taking adequate vitamin D in food resources, sun exposure, or supplementation. Special attention should be given to high-risk groups including infants, obese patients, and the elderly.

## 1. Introduction

Calciferol, generally known as vitamin D, is a fat-soluble vitamin. The human body may synthesize vitamin D with the help of sunlight, specifically ultraviolet light that falls onto the skin from a precursor of cholesterol; hence, it is a non-essential nutrient [1]. Essentially, vitamin D supplementation is not needed if we have enough sunlight exposure all year round.

Vitamin D from sunlight, food, and supplements is in an inert form; hence, it needs a biotransformation in a specific organ for activation [1]. Vitamin D is mostly synthesized endogenously when skin is exposed to ultraviolet B from sunlight which produces cholecalciferol and later is delivered to the liver by the vitamin D binding protein (VDBP). With the help of 25 hydroxylase enzymes, vitamin D undergoes hydroxylation in the liver to become 25-hydroxyvitamin D3, which is the inactive form of vitamin D. Again, the vitamin D binding protein binds to 25-hydroxyvitamin D3 and transports it to the kidney for a second hydroxylation. This activation process is carried out by the 1-α 25 hydroxylase enzyme in the kidney that forms 1,25 dihyroxyvitamin D (1,25D). Some studies also highlighted that 1,25D can be synthesized via autocrine and paracrine action by other body tissues [2]; 1,25D is the active form of vitamin D which will act on the target organ site via diffusion or an endocytic receptor for transcription [3,4].

Apart from sunlight, humans may also obtain vitamin D from food sources [5]. There are three types of sources which may increase vitamin D levels in human; they are natural foods, fortified food, as well as supplements. Vitamin D is mostly found in fatty fish such as salmon, mackerel, herring, tuna, and sardines, but it is also abundant in cod liver oil and egg yolk [1,6]. Meanwhile, vegetables, meat, and poultry are poor sources of vitamin D. For fortified food, it can be found mainly in cereals, bread, butter, yogurt, and milk, and they are mostly on a volunteer basis by the manufacturer. For supplements, vitamin D is usually combined with calcium or with a multivitamin formulation [1].

Malaysia is a tropical country with an ample source of sunshine, but, at the same time, the majority of Malaysia’s population consists of Muslims who cover themselves (especially ladies) during outdoor activities. There is also no previous review trying to look into this issue collectively to trigger Malaysian authorities regarding the silent threat of vitamin D deficiency. Thus, it is important to review the existing evidence that can shed light on the vitamin D deficiency status in Malaysia and its related causes.

This is a narrative review that only includes clinical or observational studies carried out over the past 5 years among the Malaysian population. The studies also include a measurement of serum 25 (OHD)D, and vitamin D insufficiency or deficiency were defined according to the accepted international definition. Any studies with a small number of samples that did not reflect the Malaysian population or studies that evaluated vitamin D deficiency based on clinical features without serum measurement were removed from the review.

Malaysia adapted the guidelines of IOM 2011 to be used in the Recommended Nutrient Intakes (RNI) for Malaysia. The daily requirements of vitamin D has at least doubled in RNI for Malaysia 2017 compared to the RNI for Malaysia 2005, and in following, more studies discovered that Malaysians have a higher prevalence of vitamin D deficiency than expected [7,8]. The daily requirements with the assumptions of minimal sunlight exposure are summarized in Table 1 [1].

## 2. Epidemiology of Vitamin D Deficiency

The 25-hydroxyvitamin D (25(OH)D) level is utilized to reflect dietary and endogenously acquired vitamin D. With a long half-life of 15 days, 25(OH)D can be found in plentiful concentrations in the body [5,9]. However, the accuracy of the diagnostic test and its standard references are not clearly stated and are varied according to different studies [5,9,10]. Despite the differences in the value of deficiencies, the published cut-off value is usually 30 nmol/L and often falls within the range of 25 to 30 nmol/L. The definition for insufficiency is serum 25(OH)D between 30 to 50 nmol/L [8,9]. The European Food Safety Authority (EFSA), Endocrine Society, Institute of Medicine (IOM), and the Scientific Advisory Committee on Nutrition (SACN) developed the serum 25(OH)D concentration thresholds [9].

### 2.1. Malaysia’s Prevalence

A major community-based survey and a number of selected sample studies were conducted in Malaysia to determine vitamin D status among the people. From 2004 to 2020, there were published studies of vitamin D status involving adult men and women [8,11,12,13,14,15,16], pregnant women [17,18,19], postmenopausal women [20,21], adolescents [7,22], children [23,24], and also children with specific diseases such as spina bifida [25], epilepsy [26,27], chronic liver disease [28], and atopic dermatitis [29]. The majority of the research was conducted in Peninsular Malaysia, specifically in Kuala Lumpur and tertiary hospitals. Furthermore, different studies defined and classified vitamin D inadequacy and insufficiency using different guidelines.

### 2.2. Adults

A multi-ethnic study involving male and female teachers in Kuala Lumpur, Malaysia showed that 67.4% has vitamin D deficiency (<20 ng/mL), which was comprised of Indian (80.9%), Malay (75.6%), others (44.9%), and Chinese (25.1%) [16]. According to a study among Malay men and women, 87% of women suffered vitamin D deficiency compared to 41% of men [14].

Another study in Kuala Lumpur among Malay men concluded that the prevalence of deficiency was only 0.5%, with insufficiency at 22.7%. This study involved those 20 years and above [8]. Less than one percent of women of childbearing age (18 to 40 years old) who resided in Jakarta and Malaysia had 25(OH)D deficiency, while 61% of women had otherwise shown insufficiency [11]. Another study in Kuala Lumpur discovered that Indians have the lowest mean vitamin D level. It was found that the average serum vitamin D level was 25.27 ng/mL for males and 17.24 ng/mL for females [16]. The details are summarized in Table 2.

### 2.3. Menopausal Women

Studies showed that women’s vitamin D levels are lower on average, and because of this, special populations such as menopausal women need to be highlighted. A study among Chinese postmenopausal women (menopause period more than 5 years) in two main urban cities of Malaysia revealed that the majority had serum vitamin D deficiency (82.7%) [20]. Meanwhile, a multi-ethnic (Malay and Chinese) study on postmenopausal women aged 50 to 65 years in Kuala Lumpur discovered that 26.7% of Malay and 87.8% of Chinese women had vitamin D hypovitaminosis. Furthermore, there was a substantially higher prevalence of deficiency among Malay postmenopausal women when compared to Chinese women, which were 71.3% and 12.2% respectively. It was shown that the level of 25 (OH) D among Malay women was 44.4 nmol/L, whilst the Chinese recorded a level of 68.8 nmol/L [21].

### 2.4. Pregnant Women

According to the World Health Organization, in certain populations around the world, vitamin D deficiency is considered a frequent occurrence among pregnant women, and it has been linked to an increased risk of pre-eclampsia, gestational diabetes mellitus, premature delivery, and other tissue-specific disorders. A study that enrolled first trimester women 18 to 40 years old in Selangor reported that hypovitaminosis D was 90.4 percent [17]. On the other hand, a total of 50.2% of women in their third trimester (37 weeks and above) were vitamin D deficient [18] compared to 42.6% among those who were pregnant at 28 weeks [19]. Detailed results are summarized in Table 2.

### 2.5. Adolescents

A large study involving 1361 adolescents aged 12 to 13 years old in Perak, Selangor, and Kuala Lumpur revealed that 78.9% had vitamin D deficiency; 1.5% were severely deficient; and 13.7% were having insufficiency. Only 7.4% of the individuals had an adequate level of vitamin D [7]. The same study showed that the serum vitamin D mean level for males was 37.4 nmol/L (±1.2) and for females was 24.2 nmol/L (±0.6), with the Indians having the lowest mean value. Another study in Kuala Lumpur and Selangor comprising 1061 15-year-old adolescents found that vitamin D deficiency was 33% [22]. The serum vitamin D mean level for males was 70 nmol/L (±16) and for females was 53 nmol/L (±15), with Indians and Malays having the lowest mean value (Table 3).

### 2.6. Children

The South East Asian Nutrition Survey (SEANUTS) was conducted throughout Malaysia from 2010 to 2011 (Sabah, Sarawak, and Peninsular Malaysia), encompassing 3542 children aged four to twelve years old. Blood parameters revealed that 47.5% had low vitamin D levels, with females (54.1%) having a substantially greater prevalence than boys (41.1%) [24]. In contrast, a study examining 402 children aged seven to twelve years in Kuala Lumpur discovered that 35.3% of children had vitamin D deficiency, and 37.1% of children had vitamin D insufficiency concentrations (Table 3) [23].

#### 2.6.1. Children with Epilepsy

A study in two tertiary hospitals in Malaysia among children with epilepsy and on anti-epileptic medications (AED) showed that 22.5 percent were found to have vitamin D deficiency, and 19.7 percent were having vitamin D insufficiency [26]. The same study reported polytherapy of more than one AED, age, ethnicity, and sun exposure as risk factors (Table 4).

#### 2.6.2. Children with Spina Bifida

Fong et al. (2020) [25] reported 22.5 percent cases of vitamin D deficiency among 80 children with spina bifida, aged 2 to 18 years old from tertiary hospitals in Kuala Lumpur, Melaka, and Seremban. A total of 32.5% of the children had vitamin D insufficiency status. Two measures of reduced exposure to sunlight (less than 21% of body surface area, less than 35 min per day) were both associated factors (Table 4) [25].

#### 2.6.3. Children with Chronic Liver Disease

The results of a study showed that 14 percent of the children were having vitamin D deficiency, and 14 percent were having vitamin D insufficiency [28]. Among the diagnoses involved were autoimmune hepatitis, biliary atresia, and sclerosing cholangitis. The poor vitamin D status was substantially higher in children with bilirubin levels more than or equal to 34 μmol/L (Table 4).

#### 2.6.4. Children with Atopic Dermatitis

In children with atopic dermatitis (AD) at a tertiary hospital in Malaysia, it was found that 29.5% were vitamin D deficient, and 35.5% were vitamin D insufficient. However, the 25(OH)D levels in children with AD were not significantly different from children without AD. Nevertheless, the odds of having vitamin D deficiency in children with severe AD was 3.82 times higher than that of children with non-severe AD (Table 4) [29].

## 3. Discussion

Recent statistics of global vitamin D status around the world revealed that vitamin D deficiency and insufficiency are ubiquitous regardless of the latitude of the countries, and, even in high income countries, vitamin D deficiency persists despite their capabilities of fortification efforts aimed at assuring adequate intake [36]. Unfortunately, to pinpoint the most vulnerable groups exactly in terms of geographical location or countries is quite difficult due to the lack of standardized data in many countries. The available data are derived mostly from out-of-date studies (ten years and above), and the majority involved small studies rather than large surveys. Nevertheless, evidence indicates that vitamin D deficiency is more prevalent in Africa, the Middle East, and Asia [9,37] with an inclination towards female gender, particularly in pregnant and lactating women, the elderly, and those involving the use of extensive coverage of the skin, thus limiting exposure to sunlight. Additionally, there are substantial data gaps, particularly for lower middle-income countries and those pertaining to obesity which can also contribute to the problem in this current era.

This review aims to provide an updated overview of Malaysia’s vitamin D status. At present, there is no consensus for the definition of vitamin D deficiency and insufficiency levels; thus, various definitions were used in the studies included in this review (Table 2, Table 3 and Table 4). Nevertheless, it showed that through all the ages, with or without underlying disease, female gender tends to have a lower mean level of serum 25(OH)D concentration and a higher prevalence of vitamin D deficiency. High prevalence of low vitamin D levels may be related to several issues. It is known that vitamin D is mostly obtained from sunlight [38,39]. The synthesis of vitamin D3 under the skin is affected when the transmission of solar UVB radiation to the earth’s surface or UVB radiation penetration into the skin is affected [40,41]. In this context, skin type is an important factor in which melanin is very effective at absorbing UVB rays and shields the skin underneath. However, vitamin D production in the skin is further hampered by the reduced amount of UVR that is accessible [42]. Dark skin is known to have a low capacity to produce vitamin D [34,36]. This is consistent with the studies reported in Malaysia whereby the darker skin ethnicities (Indian and Malay) have a lower vitamin D mean level and are more susceptible to vitamin D deficiency and insufficiency [7,8,16,17,21]. Furthermore, the determination of skin type using the Fitzpatrick skin type chart and Mexameter (MX 18) among first trimester pregnant mothers in Malaysia [43] found that the Indians were mostly within type V to VI (dark brown to black), followed by Malays within type III (light brown), and the Chinese within type II (white skin) which objectively explained the previous results of low levels of vitamin D among Indians and Malays.

Sun avoidant lifestyles such as the use of sunscreen, conservative clothing habits, and outdoor inactivity are also some of the important causes of low vitamin D status. A sunscreen applied topically absorbs incoming UVB light, thus reducing vitamin D3 production in the skin [44]. Type of dress which involved covering the entire skin and preventing it from being exposed to sunlight also prevents the absorption, which explains why vitamin D deficiency is so widespread even in the sunniest parts of the world [18,45,46,47]. This can be seen in populations where extensive skin coverage was practiced by the women as part of their religion or cultural norm, which is often described in studies in the Middle East, and Central and South America. A study in Malaysia among multi-ethnic pregnant women in their third trimester found that veiled clothing was significantly associated with vitamin D deficiency [18]. Furthermore, individuals who are confined to their homes or work in jobs that limit their exposure to sunshine are unlikely to receive enough vitamin D from sunlight [48].

Human milk alone is not able to provide the vitamin D requirement for infants [30] unless pregnant mothers are supplemented with a high amount of vitamin D [49]. Thus, breastfed infants are among the vulnerable groups who are at risk to have a low vitamin D status. Nevertheless, a lot of other factors need to be considered since mothers are not routinely given vitamin D supplements; exclusively and partly breastfed babies should be considered to be given 400 IU of vitamin D per day [49], which is the recommended daily requirement throughout infancy (American Association of Pediatricians (AAP)).

A reduced amount of the precursor (7-dehydrocholesterol) of vitamin D3 in the skin is also linked to aging [50] in which the skin of the elderly is unable to manufacture vitamin D as effectively as in the younger people. Other factors that made them more prone to have a low level of vitamin D are increased indoor time due to their limitations [30].

An obese person has lower serum 25(OH)D levels. They require higher vitamin D doses than normal to attain the levels equivalent to those of healthy weight individuals [30]. Since vitamin D is a fat-soluble vitamin, it is easily absorbed by fat cells; thus, there will be less of it in the circulation. For those with gastric bypass surgery, the portion of the upper small intestine where vitamin D is absorbed is skipped; thus, without enough vitamin D from the diet or supplements, this group of people is more susceptible [51,52]. This is also why those who have fat malabsorption disorder, which means a limited ability to absorb fat from the diet, are susceptible to have a deficiency of vitamin D [53]. Other diseases associated with fat malabsorption are liver illness, Crohn’s disease, and cystic fibrosis [39].

The earth receives UVB photons depending on the degree of sunshine on the planet [41,54]. This explains why vitamin D3 production is minimal throughout the winter. Even during clear skies, cities in latitudes greater than 35° will receive a limited quantity of UVB radiation during the winter months [55]. In areas where UVB radiation is abundant year-round, thick cloud cover or pollution might obstruct vitamin D production [56]. As for Malaysia, a tropical country with abundant sunshine throughout the year (latitude 3.13° N, 101.7° E), its population should have sufficient vitamin D [22]. Despite that, vitamin D deficiency is still prevalent.

Among the consequences of vitamin D deficiency are nutritional rickets which can be attributed to the lack of vitamin D or dietary calcium or both even in the absence of overt deficiency [57]. Bone mineralization is sustained by an interaction of vitamin D and calcium. When there is appropriate calcium intake, 25(OH)D more than 30 nmol/L is sufficient to avoid nutritional rickets [58]. Severe vitamin D deficiency can cause a decreased level of calcium, phosphate, or phosphorus, leading to improper mineralization of the bone. These in turn can give rise to osteomalacia in adults [59]. Babies are at risk of congenital rickets and hypocalcemia if their mothers have vitamin D deficiency [60,61]. Other than preventing congenital rickets and hypocalcemia, pregnant mothers supplemented with vitamin D will be less susceptible to pre-eclampsia, having low birth weight newborn, and preterm delivery [10]. However, pregnancy supplementation of vitamin D is still not recommended since the data available are still not conclusive enough. Trial results published until September 2017 revealed potential benefits of vitamin D supplementation such as increasing birth weight, decreasing the probability of small gestational age (SGA), increasing length at one year of age, and, up to 3 years of age, there is a lower chance of children developing asthma or chronic wheezing [62]. Nevertheless, no evidence was found associating prenatal supplementation with preterm birth. Recent evidence showed that vitamin D insufficiency is able to give rise to adverse respiratory outcomes, notably asthma exacerbations [63,64] and tuberculosis (TB) reactivation [65,66]. Several studies show the role of cofactor by vitamin D in the establishment of antimycobacterial action, whereby it is a strong modulator in the immune response [67,68]. Evidence in the form of meta-analyses showed that vitamin D supplementation manages to lower the incidence of upper respiratory infections (URIs) and exacerbations of asthma [64,69].

Vitamin D deficiency can be managed with pharmacological or non-pharmacological approaches depending on the severity. The goal of the treatment is to raise serum vitamin D to normal levels to relieve the symptoms and reduce the adverse health outcomes such as rickettsia among infants, osteomalacia among adults, and osteoporosis among the elderly. To date, there are no clear guidelines in treating vitamin D deficiency in Malaysia. According to the American Family Physician, ergocalciferol at 50,000 IU given orally is the medication of choice that should be given every week for eight weeks as an effective regiment to normalize the level of serum 25-hydroxyvitamin D. The regime shall be repeated if the minimum level of 30 ng per mL is not achieved. An 800 to 1000 IU of maintenance dose daily should be started after the ergocalciferol regime. It can be obtained through diet or supplementation. In case the serum level does not rise, non-adherence to therapy or malabsorption should be suspected [70]. Children with vitamin D deficiency should also be treated to increase their serum levels to the normal range (≥50 nmol/L). Children with features of rickets should be referred to and treated by a specialist. They may require 2000 IU daily for a minimum of 3 months, together with calcium and phosphate supplements. Long-term treatment includes education on exposure to sunlight and sufficient food intake [71].

As Malaysia lies in the equator region with plenty of sunlight, our community is predicted to obtain adequate sunlight exposure for the endogenous synthesis of vitamin D through the skin. Nevertheless, it was found that sunlight avoidance practices were found to be prominent among Malaysians. A study comparing Kuala Lumpur and Aberdeen city shows that ultraviolet B exposure among Asians who reside in Aberdeen are much higher compared to their origin counterpart in Malaysia despite ultraviolet B exposure being the major source of vitamin D in Kuala Lumpur all year long [72]. Exposure of the skin of the arm and face to sunlight without sunscreen for approximately 30 min, twice weekly, can increase serum vitamin D by 40%. People with a darker skin tone may require about a three-fold longer exposure time to make the equivalent amount of vitamin D as compared to those with fairer skin. Nevertheless, caution should be taken, as individuals with sensitive skin or over exposure to the sun heat may cause skin irritation [1,73,74].

Vitamin D can also be acquired from natural food sources such as eggs, meat, fish (mackerel, salmon, sardines), and fish oils. The best source of vitamin D comes from animal products. Vegetarians can obtain dietary vitamin D from mushrooms or other fortified products such as dairy products, bread, orange juices, and cereals [1,74]. According to the Malaysian Food Regulations 1985, margarine and processed cereal-based foods for infants and young children are the only foods directed to be fortified with vitamin D under law. In each 100 g, table margarine shall contain not less than 250 IU of vitamin D but not more than 350 IU. For processed cereal-based foods for infants and young children, the amount of vitamin D must be in the range of 1 microgram per 100 kcal to 3 micrograms per 100 kcal (0.25 μg per 100 kJ to 0.75 μg per 100 kJ) [75]. However, it is known that other food suppliers in Malaysia also voluntarily fortified other food products with vitamin D such as milk (1.3 µg/100 g) [76], yogurt (1.2 µg/100 g) [76], canned fish, bread spread, beverages, and supplements which is evidenced from a Malaysian study which found that vitamin D deficiency is profound among those with a low intake of milk and dairy products [20,77]. Malaysia is in dire need of baseline data of the complete list of fortified food with vitamin D and consumption data of these foods. These data are essential for establishing strategies to tackle inadequate intake of vitamin D among Malaysians further.

A previous study reporting on vitamin D deficiency status among third trimester pregnant mothers in Malaysia came up with a list of foods fortified with vitamin D [19]. The list was derived according to the guideline by the Ministry of Health, Malaysia [78], in which foods claiming to be fortified with vitamin D must contain at least 5% of the Nutrient Reference Value (NRV) per serving.

Vitamin D supplementation is recommended to those who are unable to obtain adequate consumption from the diet or have inadequate exposure to sunlight such as those who are homebound or the elderly population. In Malaysia, vitamin D supplements are available over the counter in tablet form (Vitamin D 1000 IU) and are also included in multivitamin supplements with varying doses by different brands, commonly ranging from 400 IU to 600 IU. Vitamin D can be acquired in utero with additional storage enough for the first month of life. However, breast milk is a poor source of vitamin D. A local study shows that mothers with low serum vitamin D produce breast milk with low vitamin D levels; hence, a minimum of 400 IU (10 µg/day) is recommended for children and adolescents, especially among exclusively breastfed infants and all children and adolescents who are not routinely exposed to sunlight [70,79]. Patients who are obese and on medications including anticonvulsant, antifungals, antiretrovirals, and glucocorticoids may require at least 2–3 times higher doses than normal populations [74]. However, there was no conclusive evidence that higher vitamin D dose intake among obese patients can be beneficial for them. There was also no recommendation to increase vitamin D uptake among pregnant and lactating mothers due to the lack of evidence of the need of higher vitamin D as compared to non-pregnant and non-lactating mothers. Vitamin D requirements are higher among the elderly due to increased bone loss, less efficient synthesis of endogenous vitamin D, and increased PTH which predispose them to falls and fractures. Meanwhile, vitamin D supplementation is contraindicated among patients with granulomatous diseases such as tuberculosis, metastatic bone cancer, sarcoidosis, and Williams syndrome [1,70]. However, caution should be taken while consuming vitamin D supplementation. Vitamin D intake of 2000 IU daily or more predisposes individuals to vitamin D toxicity. Signs that can be observed following this condition are headache, metallic taste, nephrocalcinosis or vascular calcinosis, pancreatitis, nausea, and vomiting. Tolerable upper intake levels (UL) for vitamin D according to RNI for various ages of Malaysians are stated in Table 1 [1,70].

## 4. Conclusions

Even though Malaysia is a tropical country, the population still suffers from vitamin D deficiency and insufficiency, a finding which is consistent with research from other countries. It is imperative to investigate further the causes of the high prevalence of low vitamin D levels with standardized guidelines for the vitamin D level definition, larger sample sizes, and solid methodologies. The latest evidence showed that vitamin D is important for the prevention of multiple non-communicable diseases, and it has significant roles in other organ functions. Especially during this COVID-19 pandemic, whereby the presence of underlying inflammation determined the severity of COVID-19 illness, and low levels of vitamin D are associated with a significantly increased risk of pneumonia and viral upper respiratory infections. With these in mind, there should be more effort and investment put into the empirical research to be incorporated into effective interventions to reduce the burden of vitamin D deficiency and thus other preventable diseases. Regulatory bodies should make vitamin D content in Malaysian food composition table as mandatory. In addition, the vitamin D fortification in Malaysia should be regulated and standardized to control this issue.

## Figures and Tables

**Table 1 nutrients-14-00567-t001:** Recommended nutrient intakes for Malaysia.

Age Group	RNI 2005	RNI 2005	RNI 2017	RNI 2017	Upper Limit (µg/Day)
	(µg/Day)	(IU/Day)	(µg/Day)	(IU/Day)	
Infants					
0–5 months	5	200	10	400	25
6–11 months	5	200	10	400	37.5
Children					
1–3 years	5	200	15	600	
4–6 years	5	200	15	600	100
7–9 years	5	200	15	600	
Boys					
10–18 years	5	200	15	600	100
Girls					
10–18 years	5	200	15	600	100
Men					
19–50 years	5	200	15	600	
51–65 years	10	400	15	600	100
>65 years	15	600	20	800	
Women					
19–50 years	5	200	15	600	
51–65 years	10	400	15	600	100
>65 years	15	600	20	800	
Pregnancy	5	200	15	600	100
Lactation	5	200	15	600	100

**Table 2 nutrients-14-00567-t002:** Prevalence studies among the adult age group.

Author/ Year	Setting	Sampling Frame	Sample Size	Cut off Definition Reference	Insufficiency Prevalence (%)	Deficiency Prevalence (%)	Mean Level (nmol/L)	Findings
Leiu et al., 2020 [20]	Members of 15 affiliates under the National Council of Senior Citizens Organisations Malaysia (NACSCOM) in Kuala Lumpur and Selangor	Women, menopausal for at least five years or more, aged 50 years and above	214	Insufficiency: 30–50 nmol/L Deficiency: <30 nmol/L Reference: Institute of Medicine, 2011 [30]	49.5	33.2	37.4 ± 14.3	High percentage of body fat (*p* < 0.01) and low consumption of milk and dairy products (*p* < 0.05) were the main contributors towards insufficient serum vitamin D levels, but not socio-demographic characteristics, other anthropometric indices, sun exposure, or diet quality.
Lee et al., 2020 [18]	Tertiary hospital Obstetric and Gynecology department in Selangor	Pregnant women 3rd trimester (≥37 weeks), singleton, aged 19 to 40 years	217	Deficiency: <30 nmol/L Reference: Institute of Medicine, 2011 [30]	-	50.2	29.8	Age (*p* < 0.01), veiled clothing (*p* < 0.01), maternal vitamin D intakes from both food and supplements (*p* < 0.01), and GC rs7041 (*p* < 0.05) and GC diplotypes (*p* < 0.05) significantly associated with vitamin D deficiency
Woon et al., 2019 [19]	Maternal and child health government clinic in Selangor	Multi-ethnic pregnant women 3rd trimester (≥28 weeks), singleton	535	Insufficiency: 30–50 nmol/L Deficiency: <30 nmol/L Reference: Institute of Medicine, 2011 [30]	49.3	42.6	33.8	Higher intake of vitamin D (*p* < 0.01), non- Malay ethnicity (*p* < 0.001) associated with lower odds of having vitamin D deficiency. No associations were found between age, educational level, monthly household income, work status, gravidity, parity, pre-pregnancy body mass index, total hours of sun exposure, total percentage of body surface area, or sun exposure index per day.
Shafinaz & Moy 2016 [16]	Government secondary school in Kuala Lumpur	Multi-ethnic male and female teachers	858	Deficiency: <50 nmol/L Reference: US Endocrine Society Clinical Practice Guidelines, 2011 [31]	-	80.9 (Indian) 75.6 (Malay) 25.1 (Chinese)	38.61 (Indian) 41.36 (Malay) 63.50 (Chinese)	Malays, Indians, and females; higher BMI and larger waist circumference were significantly associated (*p* < 0.05) with lower serum 25(OH)D level.
Bukhary et al., 2016 [17]	Government clinics, in Selangor	Multi-ethnic, pregnant women in 1st trimester, aged 18–40 years old	396	Deficiency: <50 nmol/L Reference: Not mentioned	-	90.4% (<50 nmol/L)	27.11 (median)	Independent predictors of hypovitaminosis D were Malay ethnicity (*p* < 0.001), Indian ethnicity (*p* < 0.001), secondary education (*p* = 0.001), and tertiary education (*p* < 0.001).
Chin et al., 2014 [8]	Health screening session of the Malaysian Aging Male Study, in Kuala Lumpur	Multi-ethnic men, 20 years and above	383	Insufficiency: 30–50 nmol/L Deficiency: <30 nmol/L Reference: Institute of Medicine, 2011 [30]	22.7	0.5	58.7	Being Chinese, being older in age, having lower body mass index (BMI), and having a high physical activity status were significantly associated (*p* < 0.05) with a higher serum 25(OH)D level.
Rahman et al., 2004 [21]	Public community (senior citizen clubs, residential areas and religious centers) in Kuala Lumpur	Chinese and Malay women, more than 5 years post-menopausal, aged between 50 and 65 years	276	Insufficiency: 25–50 nmol/L Deficiency: <25 nmol/L Reference: Not mentioned	71.3 (Malay) 12.2 (Chinese)	-	44.4 ± 10.6 (Malay) 68.8 ± 15.7 (Chinese)	Ethnicity has a strong association with vitamin D status (*p* < 0.001).
Moy & Bulgiba 2011 [14]	Public university, Kuala Lumpur	Malay ethnic, male and female employees, aged 35 years and above	380	Insufficiency: 25–49.9 nmol/L Deficiency: <25 nmol/L Reference: Not mentioned	87 (female) 41 (male)	-	36.2 (Female) 56.2 (Male)	1-year age increments (*p* = 0.007), being female (*p* < 0.001), and abdominal obesity (*p* = 0.001) significantly associated with insufficient vitamin D status. Respondents with insufficient vitamin D were found to have higher odds of having Metabolic Syndrome after adjusting for age and sex (*p* = 0.044).
Green et al., 2008 [11]	Cities in Jakarta and Kuala Lumpur	Multi-ethnic, non-pregnant women, 18–40 years old	504	Insufficiency: <50 nmol/L Deficiency: <17.5 nmol/L Reference: Not mentioned	61	0.8	48	The relation between vitamin D status and parathyroid hormone concentration did not differ between women with low, medium, or high calcium intakes (*p* = 0.611).

**Table 3 nutrients-14-00567-t003:** Prevalence studies among adolescent and children age group.

Author/Year	Setting	Sampling Frame	Sample Size	Cut off Definition Reference	Insufficiency Prevalence (%)	Deficiency Prevalence (%)	Mean Level (nmol/L)	Findings
Quah et al., 2018 [22]	Public secondary schools in central and northern Malaysia	Multi-ethnic, male, and female, aged 14–15 years old	1061	Deficiency: ≤50 nmol/L Reference: Vitamin D supplementation guidelines (Pludowski et al., 2018) [32]	-	33	53 ± 15 (Female) 70 ± 16 (Male) 69 ± 15 (Chinese) 58 ± 18 (Malay) 58± 15 (Indian)	Female (*p* < 0.001), Malay, and Indian (*p* = 0.02); those always wearing long sleeves (*p* = 0.05) were more likely to have vitamin D deficiency. For female participants, ethnicity (*p* = 0.005) was an important risk factor. Cloud cover (*p* = 0.33), school residence (*p* = 0.17), skin pigmentation (*p* = 0.36), sun-exposure (*p* > 0.05), and sun-protective behaviors (*p* > 0.05) were not significant risk factors.
Al-Sadat et al., 2016 [7]	Public secondary schools in central and northern Malaysia	Multi-ethnic, male, and female, aged 12–13 years old	1361	Insufficiency: 37.5–50 nmol/L Deficiency: <37.5 nmol/L Reference: Vitamin D Deficiency in Children and Its Management: Review of Current Knowledge and Recommendations (Misra et al., 2008) [33]	13.7	78.9%	37.4 ± 1.2 (Male) 24.2 ± 0.6 (Female) 29.1 ± 0.8 (Malay) 30.8 ± 1.8 (Chinese) 26.6 ± 1.6 (Indian) 36.1 ± 5.0 (Others)	Females (*p* < 0.001), adolescents with wider waist circumference (*p* < 0.001), and those in urban areas (*p* < 0.001) had higher risks of being vitamin D deficient.
Poh et al., 2013 [24]	Schools, kindergartens, and nurseries throughout Malaysia	Multi-ethnic, male, and female, aged 4–12 years	3542	Insufficiency: <50 nmol/L Reference: Vitamin D Deficiency in Children and Its Management: Review of Current Knowledge and Recommendations (Misra et al., 2008) [33]	47.5	-	-	High prevalence of vitamin D insufficiency and the inadequate intake of calcium and vitamin D.
Khor et al., 2011 [23]	Primary schools in Kuala Lumpur	Multi-ethnic, male, and female, aged 7–12 years old	402	Insufficiency: 37.5–50 nmol/L Deficiency: <37.5 nmol/L Reference: Vitamin D Deficiency in Children and Its Management: Review of Current Knowledge and Recommendations (Misra et al. 2008) [33]	37.1	35.3	-	Significant inverse association was found between serum vitamin D status and BMI-for-age among boys (*p* = 0.016).

**Table 4 nutrients-14-00567-t004:** Prevalence studies among children with underlying diseases.

Author/Year	Setting	Sampling Frame	Sample Size	Cut off Definition Reference	Insufficiency Prevalence (%)	Deficiency Prevalence (%)	Mean Level (nmol/L)	Findings
Fong et al., 2020 [25]	Tertiary hospitals	Children with spina bifida, aged 2–18 years old	80	Insufficiency: 37.5–50 nmol/L Deficiency: <37.5 nmol/L Reference: Vitamin D Deficiency in Children and Its Management: Review of Current Knowledge and Recommendations (Misra et al., 2008) [33]	19.7	22.5	52.8 ± 19.1	Skin exposure to sunlight ≤ 21% body surface area (*p* = 0.007) was a significant risk factor for vitamin D deficiency; duration of sun exposure ≤ 35 min/day (*p* = 0.029) was a significant risk factor for vitamin D insufficiency.
Lee et al., 2019 [28]	Tertiary hospital	Children with chronic liver disease, aged <1 to 18 years old	59	Insufficiency: 30–50 nmol/L Deficiency: ≤30 nmol/L Reference: Institute of Medicine, 2011 [30], European Society of Pediatric Gastroenterology, Hepatology and Nutrition (ESPGHAN) (Braegger et al. 2013) [34]	14	14	80.1 ± 53.5	The proportion of children with either deficient or insufficient vitamin D status was significantly higher in children with bilirubin level ≥34 mmol/L than in children <34 mmol/L (*p* = 0.028).
Lee, Choon et al., 2019 [29]	Dermatology clinic in a tertiary hospital	Children with severe atopic dermatitis (AD), aged less than 18 years old	200	Insufficiency: 50–75 nmol/L Deficiency: <50 nmol/L Reference: Vitamin D Deficiency (Holick 2007) [35]	35.5	29.5	62.9 (Median, with AD) 64.65 (Median, without AD) 39.94 (Median, severe AD) 65.64 (Median, mild-moderate AD)	The serum levels of 25(OH)D among children with AD was not statistically different from children without AD (*p* = 0.616); serum vitamin D levels were significantly lower in children with severe AD compared to those with mild-to-moderate AD (*p* = 0.021); the odds of having vitamin D deficiency in children with severe AD was 3.82 times that of children with non-severe AD (*p* < 0.05).
Fong et al., 2016 [26]	Pediatric clinic in a tertiary hospital	Children with epilepsy, ambulant, on long-term (>1 year) anti-epileptic drugs (AEDs), aged between 3 and 18 years old	244	Insufficiency: 37.5–50 nmol/L Deficiency: <37.5 nmol/L Reference: Vitamin D Deficiency in Children and Its Management: Review of Current Knowledge and Recommendations (Misra et al., 2008) [33]	19.7	22.5	53.9	Polytherapy >1 AED (*p* = 0.032), age >12 years (*p* = 0.032), Indian ethnicity (*p* < 0.05), sun exposure time 30–60 min/day (*p* = 0.038), sun exposure time <30 min/day (*p* = 0.002), and female (*p* = 0.006) are significant risk factors for vitamin D deficiency.

## Data Availability

Data sharing is not applicable to this article. No new data were created or analyzed in this study.

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
