# Peer review of "An Update on Vitamin D Deficiency Status in Malaysia"

_nutrients, 2022, doi:10.3390/nu14030567_

Round 1
Reviewer 1 Report
- The main intent of this review is the status in Malaysia. Therefore it is a systematic review of studies in that country. Delete the text which describes vitamin D actions (lines 55-111). Note a correction is needed at line 51. It is not just the kidney that produces 1,25D, but almost all tissues, the latter being paracrine/autocrine pathway for vitamin D action. Also note line 49 – remove “di” in the metabolite name.
- Combine Table 1 and Table 5 – these recommendations/UL values are same as IOM. Note that you need to describe intakes of Malaysians not just 25OHD values when discussing status.
- Figure 1 is not needed. You have one set of cut-offs that you use and it is not necessary to show how these vary around the world.
- A paragraph on global prevalence is not needed (lines 167-185) but you should be discussing other countries in the Discussion in comparison to Malaysia.
- AT line 222- separate pregnancy and postmenopausal – these do not go together. Postmenopausal women can be discussed with older adults in general as older persons need more D.
- Rewrite lines 365-370 for Malaysia specifically. What the latitude is. You do not have winter. DO you have a rainy season?
- Fortified foods should be put in a table and itemized. This is not a difficult task. A dietitian/nutritionist can be a co-author to ensure this is done. There must be some data to begin a list of D-containing foods. Similarly, what supplements are available in the country?
Reviewer 2 Report
The topic of the review is interesting and relevant. The authors must correct the tenses in the manuscript. Here are some suggestions.
With the help of 25 hydroxylase enzyme, vitamin D underwent initial hydroxylation which occurs in the liver to become 25-hydroxyvitamin D3 which is the inactive form of vitamin D.
Please correct the tenses.
Vitamin D has many roles in human body. Many evidence from the past and recent research have shown vitamin D is vital for human health.
Cite papers
Very well-established evidence showed that vitamin D plays a crucial role in maintaining serum calcium levels and ensuring adequate bone mineralization via three mechanisms
Cite papers
The second mechanism is when vitamin D levels go below normal, parathyroid hormone (PTH) will increase bone resorption by activating osteo clasts, hence mobilise calcium from bone. This mechanism will cause bone demineralisation which causes brittle and thin bone in a long run.
Correct the tense. Also cite papers.
Vitamin D has an important role in the immunomodula tory properties where it was found to suppress macrophage adhesion and migration in diabetic patients which in turn impaired atherosclerosis progression.
Cite papers
Vitamin D involves in many pathways in the prevention of many cancers as well as reducing the risk of tumor progression.
Correct the sentence
Multi-ethnic study involving male and female teachers in Kuala 207 Lumpur, Malaysia showed that 67.4% has vitamin D deficiency
Mention the range
In this context, skin type is an important factor in which melanin is very effective at absorbing UVB rays, thus, increasing the skin pigmentation and lowers vitamin D3 production significantly.
Rephrase the sentence.
Particularly in area where extensive skin coverage was practiced by the women as part of their religion or cultural norm, which often been described in studies in Middle East, and Central and 336 South America.
Incomplete sentence
Nevertheless, the RNI did not recommend a higher vitamin D dose intake among obese patients due to lack of conclusive evidence of its benefits.
It is not surprising that despite Malaysia is a tropical country, the population has vitamin D deficiency and insufficiency which is consistent with the existing and recent worldwide literature review.
Please rewrite the sentence.
In the conclusion, please mention few more means by which the problem can be taken care.
The authors have mentioned about covid-19 in conclusion, please write few lines in the text about the associations of vitamin D and covid-19.
Reviewer 3 Report
The review work presented by Zaleha Md Isa and colleagues titled: “An update of vitamin D deficiency status in Malaysia” is interesting, well written, clear, and easy to read. The topic is stimulating and therefore, it adds clustered information to the subject area of vitamin D compared with others published articles for the same country.
Please revise and format in the journal style all the manuscript, see figure 1, table 2 (and 3 not 5!) as well as reference
Round 2
Reviewer 1 Report
I have two major comments that were not satisfactory from first review:
- Figure 1 is a copyright violation as you do not indicate permission to use this figure that has been published in another journal.
- Table 5 is not satisfactory. These foods are not available and the list is incomplete for fortified foods in the USA.
Instead, this paragraph (lines 355-367) needs to be improved (see ALL CAPS comments)”
“… or other fortified products such as dairy products, bread, orange juices, and cereals (MIMS n.d.; Ministry of Health Malaysia 2017). [YOUR MINISTRY SAYS YOU HAVE THESE FOODS IN THE COUNTRY SO MAKE A TABLE AND LIST THEM] According to the Malaysian Food Regulations 1985, margarine and processed cereal-based foods for infants and young children are the only foods directed to be fortified with vitamin D under law. Table margarine shall contain not less than 250 IU of vitamin D but not more than 350 IU [IS THIS PER 100 GRAMS?] . For processed cereal-based foods for infants and young children, the amount of vitamin D must be in the range of 1 microgram per 100 kcal to 3 microgram per 100 kcal (0.25 µg per 100 kJ to 0.75 µg per 100 kJ) (Malaysian Food Act 1983). However, it is known that other food supplier in Malaysia also voluntarily fortified other food products with vitamin D especially milk[WITH HOW MUCH D?] and dairy products [WHICH PRODUCTS? AND HOW MUCH EACH? I KNOW HORLICKS IS FORTIFIED AND AVAILABLE IN YOUR COUNTRY ] which is evidenced from a Malaysian study which found that vitamin D deficiency is profound among those with low intake of milk and dairy products (Kruger et al. 2019; Lieu et al. 2020). Malaysia is in dire need of baseline data of complete list of fortified food with vitamin D [THIS MANUSCRIPT CAN START THIS LIST] and consumption data of these foods.”
